# Prevention of Blood Incompatibility Related Hemagglutination: Blocking of Antigen A on Red Blood Cells Using *In Silico* Designed Recombinant Anti-A scFv

**DOI:** 10.3390/antib13030064

**Published:** 2024-08-01

**Authors:** Saleha Hafeez, Najam Us Sahar Sadaf Zaidi

**Affiliations:** 1Atta-Ur-Rahman School of Applied Biosciences, National University of Sciences and Technology, H-12 Sector, Islamabad 44000, Pakistan; 2Pak-Austria Fachhochschule Institute of Applied Sciences and Technology, Haripur 22600, Pakistan

**Keywords:** disulfide-stabilized scFv, A-Trisaccharide, antigen blocking, hemagglutination, anti-A scFv, MD simulations, recombinant proteins, universal blood

## Abstract

Critical blood shortages plague healthcare systems, particularly in lower-income and middle-income countries. This affects patients requiring regular transfusions and creates challenges during emergencies where universal blood is vital. To address these shortages and support blood banks during emergencies, this study reports a method for increasing the compatibility of blood group A red blood cells (RBCs) by blocking surface antigen-A using anti-A single chain fragment variable (scFv). To enhance stability, the scFv was first modified with the addition of interdomain disulfide bonds. The most effective location for this modification was found to be H44-L232 of mutant-1a scFv. ScFv was then produced from *E.coli* BL21(DE3) and purified using a three-step process. Purified scFvs were then used to block maximum number of antigens-A on RBCs, and it was found that only monomers were functional, while dimers formed through incorrect domain-swapping were non-functional. These antigen-blocked RBCs displayed no clumping in hemagglutination testing with incompatible blood plasma. The dissociation constant K_D_ was found to be 0.724 μM. Antigen-blocked RBCs have the potential to be given to other blood groups during emergencies. This innovative approach could significantly increase the pool of usable blood, potentially saving countless lives.

## 1. Introduction

The established blood donation and transfusion system faces critical challenges that threaten its ability to save lives. These challenges include the constant threat of bloodborne diseases, blood type incompatibility, short shelf life of blood products, and most importantly, a persistent lack of voluntary donors [1]. In lower-income and middle-income countries (LMICs), the unavailability of compatible blood due to lack of voluntary donors presents a significant issue. Blood banks in these countries often struggle to maintain adequate supplies for emergencies. This affects patients who require regular transfusions, especially those with chronic conditions like thalassemia [2]. The COVID-19 pandemic further exacerbated these existing issues. Lockdowns restricted donor mobility and hindered the organization of blood donation camps, leading to a sharp decline in blood donations at a time when demand for blood was actually increasing [3,4].

Despite these challenges, blood transfusions are an essential component of modern medicine. Remarkable advancements have been witnessed in the past few decades in medical science, particularly in surgery. These breakthroughs have enabled doctors to perform life-saving procedures that were once unimaginable, procedures that rely heavily on the availability of safe blood. Blood transfusions are also essential in managing various diseases like severe anemia and thalassemia [5,6].

The urgency of blood typing and transfusions becomes especially critical in emergency situations involving severe bleeding from wounds. In these scenarios, timely administration of compatible blood can be the difference between life and death. It replaces lost blood volume and stabilizes the patient’s condition. However, emergencies can create a challenge when specific blood types are limited. This is where universal blood or O negative blood, becomes invaluable. Due to its compatibility with most blood types, it can be administered in situations where a patient’s specific blood type is unavailable [7,8]. But this is usually a last resort because O negative blood supplies can also run low during mass casualty events [9].

This shortage of universal blood has driven the search for alternative solutions. One promising approach involves the development of artificial blood or blood substitutes. While research is ongoing, no such product has been approved by FDA [10]. Another strategy focuses on enzymatically converting any blood type to type O. While this method holds promise, it currently requires appropriate enzymes and conditions, making it less practical [11].

Considering the problems associated with low donation rates in LMIC countries and the advantages of universal blood, this study focuses on the possibility of achieving functional mimicry of blood groups O and B by blocking surface antigen-A on A and AB RBCs respectively. These RBCs can be given to other blood groups from the ABO blood group system during emergencies. We used single chain fragment variable (scFv) of anti-A antibody as blocking fragments to block the antigen-A of blood groups A and AB. Once the antigen is blocked, recipient’s anti-A antibodies cannot interact with donor’s RBCs antigens and hence no blood incompatibility related transfusion reaction will occur (Appendix B
Figure A1). The idea of using scFvs as blocking fragment against blood group surface antigens was originally given by Mohammadi et al. (2016) [12]. In this study we first stabilized the structure of scFv and predicted the interactions of scFv with antigen-A through Molecular dynamics (MD) simulations. We then selected the scFv with stable interactions and produced it from *Escherichia coli* BL21(DE3) through molecular cloning technique. All antigens-A were blocked using scFv, and RBCs were tested for hemagglutination reaction. In the end we purified the functional scFvs and determined the dissociation constant K_D_ of scFv using enzyme-linked immunosorbent assay (ELISA).

## 2. Materials and Methods

### 2.1. Chemicals 

All the chemicals and reagent kits used in this study were of analytical grade and used as it is without further purification. Fresh screened A+ and AB+ blood was provided by NUST ASAB Diagnostics Lab, Islamabad, Pakistan. 

### 2.2. Modelling and Molecular Dynamics Simulations of Anti-A scFvs

Sequence of crystal structure of anti-A fragment variable Fv (PDB ID: 1JV5) was obtained from Protein Data Bank (PDB) [13]. Sequence of anti-A Fv was manually modified as follows: (1) Mutated at two sites of interface to contain at least one interdomain disulfide bond (Appendix B
Figure A2), (2) addition of (G_4_S)_3_ flexible linker to join VH and VL (VH-linker-VL) and (3) a 6x His-Tag at C-terminal. These were named as: Native (no interdomain disulfide bond), Mutant-1a (interdomain disulfide bond at interface 1) and Mutant-1b (interdomain disulfide bond at interface 2) scFvs. The location of these mutation sites was selected on the basis of domains proximity (sites must be far from Complementarity-Determining Regions (CDRs) and sites must be close enough to allow the formation of disulfide bonds) (Appendix B
Figure A3) as described previously [14]. Structures of all three scFvs were modelled by I-TASSER web server [15]. Protonation state of all three scFvs was set at pH 7.4 using web server PlayMolecule [16]. MD simulations were performed to check the stability of scFvs. These simulations were performed under physiological conditions of blood (Temperature 310 K, NaCl 154 mM and pH 7.4) for 200 ns using Amber ff99SB forcefield on GROMACS 2021.3 [17]. 

### 2.3. Molecular Docking and Molecular Dynamics Simulations

Structure of antigen-A (A-Trisaccharide) was made by using carbohydrate builder of Glycam web server [18]. Molecular Docking was performed on AutoDock tools version 1.5.6 [19] to study the interaction of a carbohydrate antigen and a protein scFv. Best docked pose was selected on the basis of three criteria: (1) Location of docked antigen (antigen should bind at CDRs), (2) conformation of antigen (both fucose and N-acetylgalactosamine should be interacting), and (3) binding energy. MD simulations were performed to check the stability of interactions formed between carbohydrate antigen and scFv under physiological conditions of blood (Temperature 310 K and NaCl 154 mM and pH 7.4). Simulations were performed on GROMACS 2021.3 for 100 ns using Amber ff99SB forcefield for scFv and Glycam-06h forcefield for carbohydrate was generated by AmberTools21 [20]. Results obtained from simulations were visualized and analyzed through Biovia Discovery Studio Visualizer 2021 [21]. Among three scFvs the one with stable interactions was selected for production. 

### 2.4. Designing and Transformation of Recombinant scFv-plasmid pET-28a (+)

Design of recombinant scFv-plasmid pET-28a (+) is given in Figure 1 in detail. Recombinant scFv construct (830 bp) was inserted between Ncol and Xhol restriction sites. To prevent frameshifting 2 additional nucleotides guanine and adenine were added. Recombinant scFv-plasmid pET-28a (+) was commercially obtained from Twist Biosciences-USA and transformed into the competent cells of *E.coli* BL21(DE3) through the traditional heat-shock method [22]. Successful transformants were screened on LB media plates containing kanamycin (50 μg/mL) and glucose (1%) after overnight incubation at 37 °C.

### 2.5. Expression and Purification of Recombinant Anti-A scFv

A 100 mL overnight culture of *E.coli* BL21(DE3) was used to inoculate 5 L of LB broth containing kanamycin and glucose (25 μg/mL and 1% respectively) and incubated at 37 °C with constant shaking for 3 h or when the OD_600_ reached 0.5. Recombinant anti-A scFv was expressed for 3 to 4 h using 1 mM IPTG incubated at 37 °C with constant shaking. Cells were harvested through centrifugation at 8217× *g* and stored in 15 mL resuspension buffer (20 mM Tris, 200 mM NaCl, pH 8) at 4 °C for 24 h. Next day cells were placed on ice for 2 h and disrupted using probe sonicator with 30 s on and 20 s off cycle at amplitude gradually increasing from 60% to 100% for 15 min. Cell debris was removed by centrifugation at 21,000× *g* for 30 min at room temperature. Cell lysate was syringe filtered through 0.45 μm filter. Recombinant proteins were purified at room temperature by nickel affinity chromatography through batch method. For purification 600 μL HisPur Ni-NTA resin was added to cell lysate and incubated at 18 °C for 16 h under constant shaking at 180 rpm. After 16 h Ni-NTA resin was washed with 2 mL resuspension buffer several times to remove any unbound protein. Wash buffer (20 mM Tris, 200 mM NaCl, 20 mM Imidazole, pH 8) was used several times to remove any weakly attached protein to the resin. Each time the resin was centrifuged at 5478× *g* and previous buffer was replaced with a new buffer. Finally, resin was incubated in 600 μL elution buffer (20 mM Tris, 200 mM NaCl, 250 mM Imidazole, pH 7.5) for 30 min at room temperature to remove all the attached recombinant protein. During each step of washing, supernatant was collected for protein quantification through A_280_ method and analyzed through non-reducing SDS-PAGE [23]. Multimers (trimers, dimers, and monomers) were separated at room temperature by size exclusion chromatography through gravity-flow method. For separation 600 μL of Ni-NTA purified sample was added to a 21 mL Sephadex G-100 (Sigma-Adrich, St. Louis, MO, USA) resin packed in an open glass column tube (10 mm × 300 mm) (GL Sciences, Shinjuku, Japan) that was already equilibrated with an equilibration buffer (10 mM Tris, 200 mM NaCl). Different fractions of 600 μL were collected and analyzed through A_280_ method (formula mentioned in Appendix B
Figure A4) and non-reducing SDS-PAGE. Final product was concentrated to half the volume (300 μL) and dialyzed against Tris-buffered saline TBS (10 mM Tris, 154 mM NaCl, pH 7.4) using cellulose regenerated dialysis tubing (MWCO 8000–14,000 Da). Final concentrated product was separated into aliquots (100 μL) and stored at room temperature.

In another experiment further purification of monomers was performed using Amicon Ultra-0.5 Centrifugal Filter (30 kDa MWCO). About 50 μL of purified SEC product was taken in centrifugal filters and centrifuged at 14,000× *g* for 5 min. Filtrate was separated and concentrate was recovered by placing the filter unit upside down in microcentrifuge collection tube and spinning for 5 min at 1000× *g*. Non-reducing SDS-PAGE was performed for analysis.

### 2.6. Blocking of Antigen-A on RBCs of Blood Groups A+ and AB+ with Recombinant Anti-A scFv

To facilitate easier removal of supernatant, a 1:2 ratio of RBCs to scFv solution was used compared to a 1:1 ratio. Packed RBCs of blood groups A+ and AB+ were separated via centrifugation at 6037× *g* and washed with TBS to remove plasma proteins. After removing TBS, 50 μL of RBCs were incubated with 100 μL of purified anti-A scFv for 5 min with constant shaking. After 5 min of incubation, supernatant was separated, and RBCs were washed repeatedly with TBS to remove any weakly bound scFv. This whole process (RBC incubation in scFv solution, followed by washing with TBS) was repeated until no further decrease in scFv concentration was detected in the supernatant, indicating blocking of the maximum number of antigens-A on the RBC surface by anti-A scFvs. As a control B+ RBCs were also coated with anti-A scFv using the same method. Presence of unbound scFv was detected using A280 method and non-reducing SDS-PAGE.

### 2.7. Detection of the Presence of Recombinant Anti-A scFv on RBCs and in Supernatant

To assess the effectiveness of antigen-A blocking, blood plasma of blood group B+ containing natural anti-A antibodies was incubated with antigen-blocked RBCs for 1 h at 37 °C with constant shaking. Following incubation, the supernatant and RBCs were separated into Eppendorf tubes. RBCs were lysed with distilled water to release any bound scFvs. To both the supernatant and lysed RBC fractions, Ni-NTA resin was added to purify scFvs (protocol mentioned in Section 2.5). Western blotting was then performed using primary rabbit anti-histidine tag polyclonal antibody (RRID:AB_10786151) and secondary HRP-linked caprine anti-rabbit IgG polyclonal antibody (RRID:AB_2099233) to detect the presence of purified scFvs. 

Additionally, hemagglutination assay was performed on the antigen-A blocked RBCs. Briefly, 10 μL aliquots of A+ and AB+ RBCs were incubated with anti-A and anti-B IgM antibodies and B+ plasma. Hemagglutination would indicate a lack of effective antigen-A blocking.

### 2.8. Purification of Functional Anti-A scFvs and Determination of Dissociation Constant K_D_

Functional anti-A scFvs were purified from antigen-blocked RBCs by repeating the experiments mentioned in Section 2.6 and Section 2.7, i.e., incubating RBCs with purified scFv solution in ratio 1:2 (method mentioned in Section 2.6), followed by hemolysis (method mentioned in Section 2.7) and purification of scFv through Ni-NTA chromatography (method mentioned in Section 2.5). Purified anti-A scFv were analyzed through non-reducing SDS-PAGE. After purification, scFv were dialyzed against TBS and lyophilized for further use.

#### Direct Ligand-Receptor Interaction Assay (LRA)

An ELISA-based direct LRA was performed in triplicates to determine the dissociation constant K_D_ of anti-A scFv, as described by Syedbasha et al. (2016) [24]. Lyophilized functional anti-A scFvs were resuspended in TBS and a two-fold serial dilution series was prepared, ranging from 1 nM to 500 μM. About 1 μL of A+ (5.1 × 10^6^ cells) RBCs were resuspended in 100 μL TBS. For details on the method used to determine the optimal concentration of RBCs, refer to Appendix A. ELISA plates were coated with RBC suspension using 0.3% glutaraldehyde as fixative solution as mentioned by [25]. Naked RBCs were used as negative control in separate wells. After coating, hemoglobin was removed from fixed RBCs by gradually exposing RBCs to hypotonic solutions of TBS (0.7% 0.5%, 0.2% and 0.09%). Fixed RBC were washed three times with wash buffer (50 mM Tris, 138 mM NaCl, 2.7 mM KCl, 0.05% Tween 20, pH 8) gently. Blocking step was performed by incubating 200 μL of 5% BSA at room temperature for 2 h. After incubation, wells were washed with wash buffer. In each well 100 μL of each scFv dilution was added and incubated for 1 h at 37 °C. Following incubation, wells were gently washed with wash buffer three times. About 100 μL of diluted (1:1000) primary anti-histidine antibody was added in each well and incubated at room temperature for 2 h. Unbound primary antibody was removed by washing with wash buffer. In each well 100 μL of diluted (1:10,000) secondary HRP-linked antibody was added and plates were incubated at room temperature for an hour. Finally, plates were washed to remove secondary antibody and TMB (3,3′,5,5′-Tetramethylbenzidine) substrate was added for 15 min. Reaction was stopped with stopping solution (0.5 M H_2_SO_4_). Absorbance was measured at 450 nm using a microtiter plate reader. Descriptive statistics were used to calculate the means X and standard deviation SD of the OD values of negative control RBCs. A cutoff values were then established using the formula X + 3(SD). Finally, the dissociation constant K_D_ was determined by nonlinear regression curve fit using GraphPad Prism [26].

## 3. Results

### 3.1. Structure Stability Analysis of scFvs

Unlike whole antibody, scFvs are mostly unstable and need to be stabilized through the addition of disulfide bonds [14]. On the basis of domain proximity two potential amino acid pairs were selected that were far from CDRs and had smallest Cα-Cα distances. These amino acid pairs were H44-L232 of interface 1 and H110-L175 of interface 2 and showed the Cα-Cα distance of 5.8 Å and 5.6 Å respectively. Distances between Cα-Cα of other amino acids H43-L233 and H111-L174 were 6.2 Å and 7.4 Å of interfaces 1 and 2 respectively. Modelled structures of both mutants (Figure 2A) showed similar structure of native scFv as shown in Appendix B
Figure A5.

The effect of mutations on protein stability was evaluated using molecular dynamics (MD) simulations. Root Mean Square Deviation (RMSD) reflects a protein’s deviation from its initial structure, with lower values indicating greater stability. In the RMSD plot (Figure 3A), all anti-A scFvs (native and mutants) remained below 4 Å throughout the 200 ns simulations under physiological conditions, suggesting overall stability. However, mutants-1a and 1b displayed lower fluctuations (maintained below 3 Å) compared to the native scFv, indicating potentially enhanced stability.

The root mean square fluctuation (RMSF) shows the flexibility of individual amino acid. In an RMSF plot (Figure 2B) of native and mutant-1b scFvs it was observed that amino acid serine (SER-131) at the end of (G_4_S)_3_ linker was less fluctuating compared to rest of the linker. Trajectory analysis (Figure 3A,B) of amino acids showed the interaction of serine (SER-131) with tryptophan (TRP-228) and proline (PRO-227) of CDR-3 region of VL chain of native and mutant-1b scFvs respectively. No such interactions were observed in mutant-1a scFv.

Further trajectory analysis revealed the distances between amino acids of framework at both interfaces. It was observed that the distance between Cα-atoms of amino acids TRP-47 and threonine (THR-229) at interface 1 of mutant-1a was small compared to mutant-1b and native scFv (Figure 2C). In contrast, the distance between Cα-atoms of alanine (ALA-106) and leucine (LEU-178) amino acids at interface 2 was small in case of both mutants compared to distance between amino acids of native scFv (Figure 2D).

Snapshots taken from MD trajectory at 0 ns (Figure 3C) and 200 ns (Figure 3D) show the positions of interacting amino acids of all six CDRs. These amino acids were selected for analysis on the basis of docking results i.e., amino acids from each CDR that were common in all selected poses of all three anti-A scFv were analyzed (Appendix B
Table A1). As shown in Figure 3C, amino acids of all three scFvs occupied the same location at 0 ns. However, at 200 ns, the amino acids of native scFv moved away from the binding site while the amino acids of mutants moved towards the binding site (Figure 3D). In contrast to mutant-1b, amino acids of mutant-1a were closer to the binding site. Shift in positions was noticeable for the TRP-228 amino acid in all three scFvs. It was observed that in native scFv, TRP-228 was at the interface 1 of VH-VL of scFv. In mutant-1a it was at the binding site of scFv and in mutant-1b it was closer to the interface 1 of VH-VL.

### 3.2. Functional Analysis of scFvs

In order for scFvs to be useful these scFv must be able to make a strong and stable interaction with an antigen. In our case the antigen is a carbohydrate (A-Trisaccharide) moiety of a glycoprotein. Molecular docking studies were carried out to predict the interactions of anti-A scFvs with antigen-A. Results of molecular docking of all three scFv are summarized in Table 1. Best poses were selected on the basis of the criteria as described above for MD simulations. These poses are 4th, 1st, and 7th of native, mutant-1a and mutant-1b respectively. In selected poses, all three saccharides were found interacting with CDRs. Interacting residues are mentioned in Appendix B
Table A1. Results of MD simulations were analyzed through MD trajectories and number of H-bonds maintained. As shown in Figure 4B, the complex of native scFv exhibited a brief dissociation of the antigen before 50 ns, followed by reattachment to the framework by 75 ns. Mutant-1a scFv showed promising results as the antigen maintained the long-term interactions till the end of 100 ns simulation (Figure 4C). In the case of mutant-1b scFv the antigen dissociated before 80 ns and while attempting to reassociate throughout the simulation, failed to establish a lasting bond (Figure 4D).

### 3.3. Expression and Purification of Mutant-1a scFv

Mutant-1a scFv was selected for production as recombinant periplasmic protein from *E. coli* BL21(DE3). Reducing environment of cytoplasm is not an ideal environment for the expression of proteins consisting of disulfide bonds as it can lead to improper protein folding [27]. Dimer, trimer, and larger multimers are often formed by scFvs during production due to short linker [28] (<15 amino acids) and repetitive linker [29] used between light (VL) and heavy (VH) chains. In our case monomers (25 kDa), dimers (50 kDa), trimers (75 kDa) and multimeric complexes (+100 kDa) were observed in elution fraction EfA1 after purification through nickel affinity chromatography (Figure 5A). Size exclusion chromatography is performed to separate the proteins according to the sizes. Larger proteins (trimer and multimeric complexes) separated early as shown in fractions F13 and F24–F28 (Figure 5B). No separation was observed in the case of 50 kDa dimers and 25 kDa monomers as shown in fractions F29–F37. Graphs showing the purification of recombinant protein through nickel affinity and size exclusion chromatography are shown in Appendix B
Figure A7. Overall concentration of recombinant proteins purified after affinity chromatography was 4.86 mg/L (Multi: 1.56 mg, Tri: 1.9 mg, Di: 0.6 mg and Mono: 0.8 mg). The concentration of monomers and dimers purified after size exclusion chromatography was 0.73 mg/L and 0.39 mg/L respectively.

Amicon Ultra-0.5 centrifugal filters were used to further purify the monomers. Here 30 kDa MWCO pore size membrane was used to recover monomers as filtrate (passed through filter unit). No separation of monomers and dimers was observed in filtrate fraction F while concentrate fraction C showed the retention of some dimers as shown in Figure 5C.

### 3.4. Blocking of Antigen-A on RBCs of Blood Groups A+ and AB+ with Recombinant anti-A scFv

To prevent blood group incompatibility related hemagglutination reaction, all antigens (antigen-A) on RBCs must be completely blocked. Figure 6A,B demonstrate the time course of scFv depletion from the supernatant. In aliquot 1 a significant decrease in scFv concentration was observed for both A+ and AB+ RBCs within 5 min. This decrease continued until 15 min, reaching a plateau that remained stable for up to 30 min. Aliquot 2 displayed a minor decrease in scFv concentration till 15 min for A+ RBCs and 10 min for AB+ RBCs. Aliquot 3 exhibited no decrease in scFv concentration, indicating the blocking of maximum number of antigens. As expected, in control B+ RBCs, anti-A scFv did not bind at all as shown in Appendix B
Figure A8.

SDS-PAGE (Figure 7A) was performed to check the type of scFv (dimer or monomer) interacting with the antigen. SDS-PAGE of aliquot 1 shows a sudden decrease in the concentration of monomers (25 kDa) in first 5 min in both cases (A+ and AB+) and after that no further decrease in concentration was observed. Faded bands of monomers were still visible from 10 to 30 min in both A+ and AB+ RBCs. No changes in concentrations of dimer (50 kDa) were observed. The gradual decrease in concentration of monomers and absence of observable change suggest that only monomers were interacting with surface antigens-A.

### 3.5. Detection of the Presence of Recombinant Anti-A scFv on RBCs and in Supernatant

In order to check the stability of interaction between mutant-1a scFv and antigen-A, antigen-blocked RBCs were incubated in B+ blood plasma containing anti-A antibodies. Following an hour incubation at 37 °C, no scFv was detectable in the supernatant (blood plasma) for either A+ or AB+ RBCs (Figure 7B). Hemolysis of RBCs shows the presence of attached monomeric form of scFvs. as shown in lanes 4 and 6. Blood incompatibility related hemagglutination reactions show no hemagglutination of antigen-blocked RBCs in the presence of anti-A IgM antibodies (wells 2 and 4) and blood plasma of B+ group (wells 8 and 10) as shown in Figure 8.

### 3.6. Purification of Functional Anti-A scFvs and Determination of Dissociation Constant K_D_


The dissociation constant K_D_ of the scFv was determined after confirming that only monomers of anti-A scFv were functional while dimers were non-functional. For this purpose, attached scFvs were purified from antigen-blocked RBCs. Non-reducing SDS-PAGE (Figure 9A) shows functional anti-A scFv (lane 4) after purification through Ni-NTA chromatography.

An ELISA-based direct LRA was performed to determine the dissociation constant K_D_ of anti-A scFv. Prior to the assay, hemoglobin was removed from RBCs to eliminate potential interference with the subsequent TMB substrate reaction. The results of assay indicated a K_D_ of 0.724 μM for the anti-A scFv, suggesting moderate affinity (Figure 9B). 

## 4. Discussion

This study investigated the possibility of making RBCs of one blood group mimic RBCs of other groups by blocking surface antigen-A, aiming to address critical blood shortages. The approach involved blocking the blood group A antigen using disulfide bond-stabilized anti-A scFvs. A blocking antibody as defined by the National Institutes of Health is an antibody which does not initiate an immune response and prevents other antibodies from interacting with antigen [30]. To prevent blood incompatibility-related hemagglutination reactions, an ideal blocking antibody should not itself cause hemagglutination by binding to the same antigens on different RBCs. In this study, anti-A scFvs were chosen as blocking fragments instead of whole antibodies due to their single antigen-binding site (monovalent). Whole antibodies, with their multiple binding sites (multivalent), are more prone to causing hemagglutination. While scFvs offer advantages like small size and ease of production compared to whole antibodies, they are inherently unstable and have lower binding affinities. This instability arises from weak interdomain interactions, which the addition of a disulfide bond helps to overcome [14]. 

In this study the addition of a disulfide bond not only enhanced the stability of both mutants, but also significantly reduced the interdomain (VL and VH) distance at both interfaces. Interestingly, placing the bond at interface 2 (mutant-1b anti-A scFv) resulted in a smaller distance reduction at interface 1, potentially compromising binding ability. This reduction in binding ability could be due to two factors: (1) interaction between the GS linker and PRO-227 amino acid of the VL CDR-3 region, and (2) the positioning of the TRP-228 amino acid of same region. Neither of these observations were found in mutant-1a anti-A scFv. The greater distance at interface 1 in mutant-1b (blue) compared to mutant-1a (red) (Figure 2C) might explain the positioning of both amino acids. This increased space likely allowed both amino acids to move outward and potentially allowed PRO-227 to come into close proximity and interact with the GS linker (Figure 3D). Conversely, placing the disulfide bond at interface 1 (mutant-1a) might have confined TRP-228 to a central location within the binding site, facilitating a stronger interaction with antigen-A. These findings suggest that in a general format (VH-linker-VL) of scFv the location of interdomain disulfide bond affects the positioning of amino acids in the CDR region, potentially affecting binding ability. Various research groups have reported the ideal location of interdomain disulfide bond and its effect on binding affinities [14,29]. The location H44-L100 (in our case H44-L232 at interface 1) is now considered as the universal site for disulfide bond in most scFvs [31]. 

The length of the peptide linker is a key determinant of scFv multimeric form [28]. Linkers exceeding 12 amino acids typically promote the formation of functional, monomeric scFvs. In our study, despite using a 15 amino acids linker, various soluble multimeric forms were observed. This could be attributed to the repetitive nature of the linker (G4S)3 The effect of repetitive and non-repetitive linkers on multimeric forms of scFvs has been reported by Arslan et al. (2022) [29]. Their findings demonstrate that repetitive linkers indeed increase the formation of various multimeric scFvs.

In this study, we observed incomplete separation of monomers from dimers during purification. This could be due to the flexible GS linker, which allowed the dimers to adopt various shapes and easily pass through SEC resin and 30 kDa MWCO filter membrane (Appendix B
Figure A9).

For successful transfusion in incompatible recipients, antigen-blocked RBCs must prevent hemagglutination i.e., these RBCs must mimic the recipient’s RBCs. This requires efficient blocking of blood group antigens on the RBC surface. The surface of RBC contains millions of antigens [32], and it is very difficult to know the exact number of specific antigens present on RBCs in a given sample. For this we employed a strategy of repeated interaction between RBCs and anti-A scFvs. This approach ensured complete coverage of antigens. Interestingly, only monomeric scFvs were observed to interact with the antigens, suggesting that only monomers were functionally active for blocking antigen-A, while dimers were not. A possible explanation for this lies in the orientation in which domain-swapping took place (Appendix B
Figure A10). During domain swapping, the VH domain of one scFv can pair with the VL domain of another in the opposite orientation. This mispairing disrupts the structure of the antigen-binding site, rendering dimers non-functional. Domain swapping has been studied in detail by various research groups [33,34]. Domain swapping has been observed by Kipriyanov et al. (2003) [35] during production. They observed that all domain swapped (scFv)2 variants were half-functional. Faded visible bands of monomers (Figure 7A) suggest that either some of the monomers were non-functional or the concentration of monomers was too low (diluted) to interact with the remaining antigens on RBCs.

The first environment that antigen-blocked RBCs will meet is the environment of the recipient’s blood which is a mixture of cells, protein, salts, and various molecules. These components, along with physiological conditions can dissociate scFvs from the antigens. To assess the strength of interaction we incubated the antigen-blocked RBCs in incompatible blood plasma for an hour, mimicking this challenged environment. Notably, no scFvs were detected in the supernatant (Figure 7B). This absence suggests that the scFv-antigen interaction is sufficiently strong to withstand the physiological conditions of blood. This result was also supported by the blood incompatibility testing (Figure 8).

The binding affinity is the strength of an interaction. It is experimentally expressed as the dissociation constant K_D_ which determines whether the molecules will form a stable complex in solution or not. Generally, lower the K_D_ value the higher the affinity of the antibody [36]. The scFv used in this study is the fragment variable Fv region of IgM anti-A antibody (AC1001) [37]. While IgM antibodies exhibit high avidity, their binding affinities are generally lower than those of IgG antibodies. This is especially true for antibodies targeting carbohydrates, because their simpler structure provides fewer opportunities to make a strong interaction, and thus tend to have lower affinity compared to antibodies targeting proteins. Even affinity-matured IgG antibodies targeting glycans have dissociation constants K_D_ in range of 100 nM to 100 μM [38]. Given the carbohydrate antigen and the IgM origin of the scFv, it was expected that monomeric mutant-1a anti-A scFv would have a lower affinity i.e., dissociation constant K_D_ in the upper nanomolar (nM) or lower micromolar (μM) range. For this reason, a broad range of dilutions were used. As expected, the ELISA-based direct LRA confirmed a moderate affinity of the monomeric mutant-1a anti-A scFv, with a dissociation constant K_D_ in the upper nanomolar (nM) range. Another research group found the dissociation constant K_D_ of anti-A wild type scFv (native scFv in our research) by surface plasmon resonance (SPR) in the lower micromolar (μM) [13].

## 5. Conclusions

This study explored a method to make the RBCs of one blood group function like the RBCs of other blood groups by blocking surface antigen-A. The antigen-blocked RBCs could work as other blood group RBCs provided that scFvs are structurally stable, do not detach from the antigen and have the same binding affinity as in the case of whole antibody. While our findings are promising, some limitations exist: (1) the computational studies were conducted under physiological conditions of blood only. In reality a blood is a very complex mixture of cells, proteins, and various salts. (2) The storage conditions of blood (blood stored in blood banks before transfusion) were not applied. These storage conditions include variations in pH and temperature. (3) The type of monomeric scFv produced (with or without interdomain disulfide bond) was not identified. (4) During washing steps many lower affinity scFvs were likely removed, leaving only comparatively high affinity scFvs bound for detection. (5) Information about the number of antigen-A molecules present on a single RBC was not known; hence, the saturation ratio between the antigen and scFv was not established.

Future research should investigate scFv behavior under more realistic conditions, including temperature and pH fluctuations, optimize linker design to ensure a higher yield of functional, monomeric scFvs for successful antigen blocking, detect the presence of interdomain disulfide bond using various techniques such as mass spectrometry, X-ray crystallography and NMR spectroscopy, focus on real-time monitoring of interactions of molecules by SPR, and computationally increase the affinity of anti-A scFv by mutating each amino acid in the CDRs and selecting the mutant with the predicted highest affinity.

## Figures and Tables

**Figure 1 antibodies-13-00064-f001:**
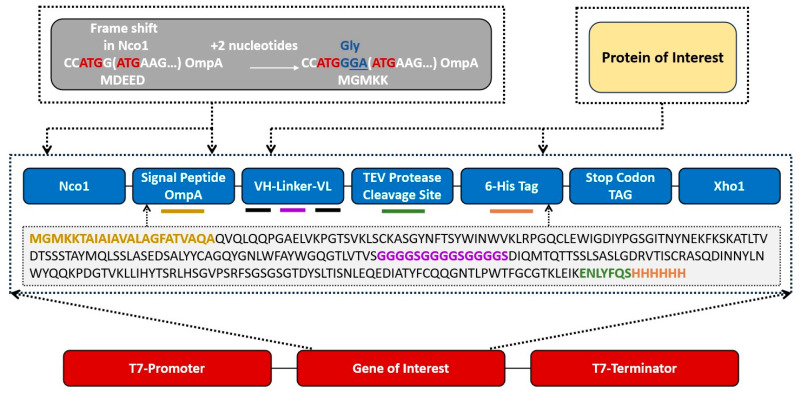
Schematic diagram of the design of mutant–1a anti–A scFv gene.

**Figure 2 antibodies-13-00064-f002:**
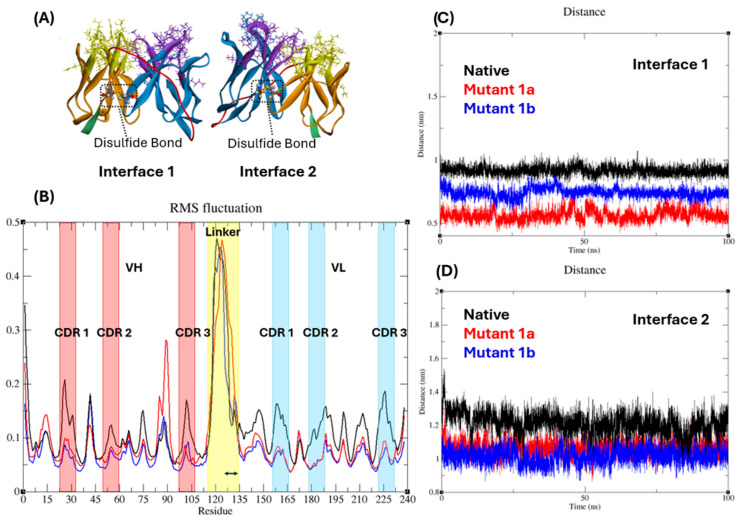
(**A**) I-TASSER models of mutants-1a (interface 1) and 1b (interface 2) scFvs. Information about the models is provided in Appendix B
Figure A5 (**B**) RMSF plot of amino acids. Double arrow represents the interacting amino acid of GS linker, black, red, and blue colors represent native, mutant-1a and mutant-1b scFvs respectively. (**C**) Distance between Cα-atoms of amino acids TRP-47 and THR-229 at interface 1 and (**D**) distance between Cα-atoms of amino acids ALA-106 and LEU-178 at interface 2.

**Figure 3 antibodies-13-00064-f003:**
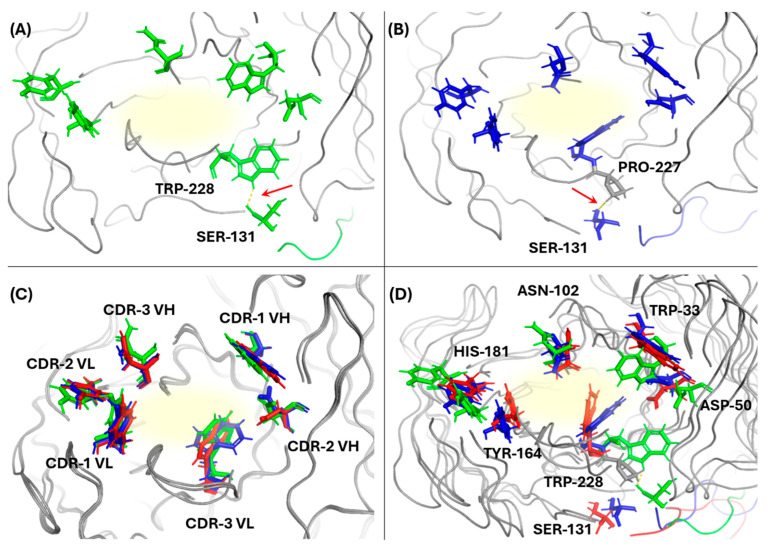
Snapshots of MD trajectories. Green, red, and blue colors represent amino acids of native, mutant-1a and mutant-1b scFvs respectively. Yellow color represents the binding center and red arrows represent the bond formed between amino acids. (**A**) Bond formed between TRP-228 of CDR-3 of VL and SER-131 of GS linker in native scFv. (**B**) Bond formed between PRO-227 (Grey color) of CDR-3 of VL and SER-131 of GS linker in mutant-1b scFv. (**C**) Snapshot showing the initial positions of interacting amino acids of CDRs at 0 ns. (**D**) Snapshot showing the final positions of interacting amino acids of CDRs at 200 ns.

**Figure 4 antibodies-13-00064-f004:**
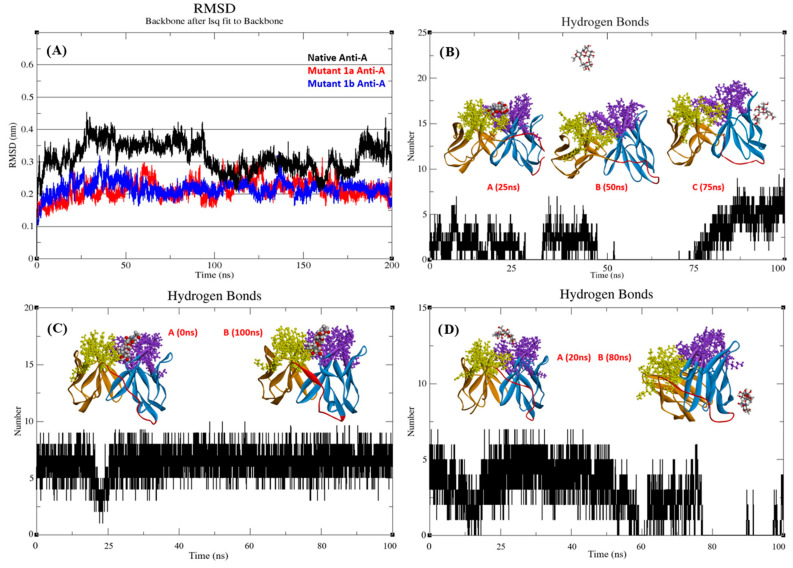
(**A**) RMSD vs. time plot of each scFv obtained along 200 ns simulation. Number of hydrogen bonds formed between antigen-A and scFv. (**B**) Native scFv, (**C**) Mutant-1a scFv and (**D**) Mutant-1b scFv. High resolution images are shown in Appendix B
Figure A6A–D.

**Figure 5 antibodies-13-00064-f005:**
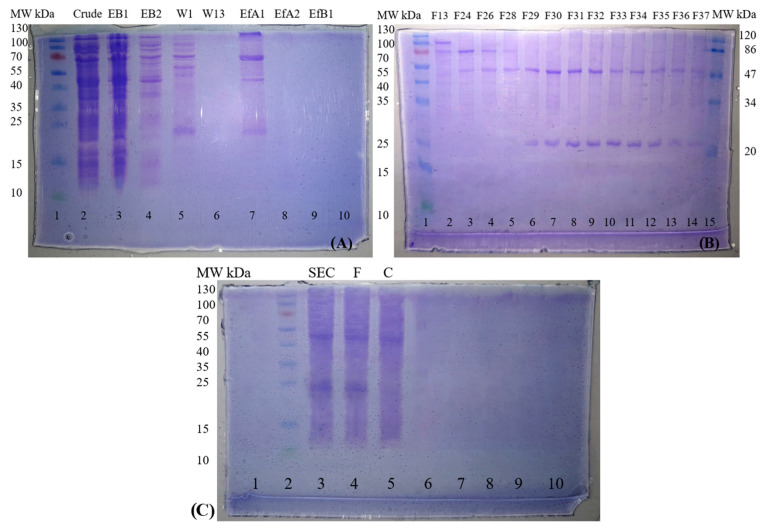
(**A**) Purification of mutant-1a anti-A scFv through nickel affinity chromatography. Coomassie R-250 stained 15% SDS PAGE gel showing purification of recombinant protein before (lanes 2–6) and after elution (lanes 7–9). EfA1 shows 4 bands of ~25 kDa, 50 kDa, 75 kDa and +100 kDa. (**B**) Separation of recombinant multimers on the basis of molecular weight through size exclusion chromatography. (**C**) Separation of monomers and dimers through centrifugal filters (30 kDa MWCO). Lane 3 shows the purified product (monomers and dimers) of size exclusion chromatography. Lanes 4 and 5 shows the recombinant protein recovered as filtrate (F) and concentrate (C) respectively.

**Figure 6 antibodies-13-00064-f006:**
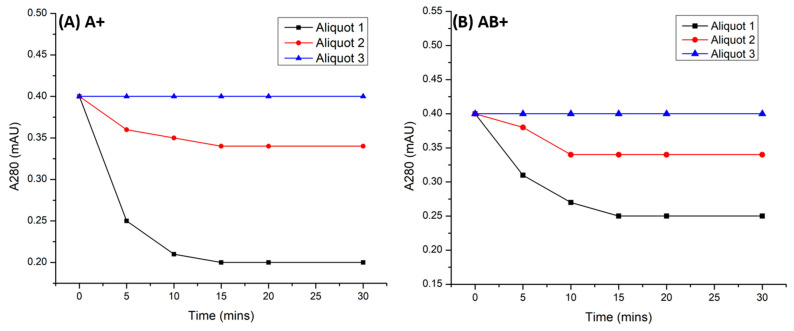
Graphs (**A**,**B**) show the gradual decrease in the concentrations of mutant-1a anti-A scFv in the supernatant containing A+ and AB+ RBCs. Each aliquot represents a portion of a larger sample with the same protein concentration for analysis. Each aliquot contains 0.73 mg/L and 0.39 mg/L of SEC purified monomeric and dimeric anti-A scFv respectively.

**Figure 7 antibodies-13-00064-f007:**
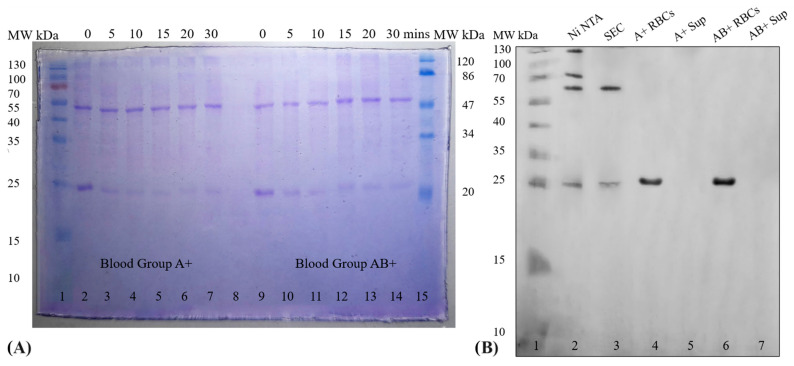
(**A**) Coomassie R-250 stained 15% SDS PAGE gel shows decrease in concentration of scFv monomer (25 kDa) from 0 to 30 min (lanes 2–7 in the case of A+ and lanes 9–14 in the case of AB+). (**B**) Western blot showing all histidine tagged proteins: Ni-NTA purified proteins (lane 2), size exclusion chromatography separated proteins (lane 3), scFvs attached on A+ and AB+ RBCs (lanes 4 and 6) and scFvs in supernatant of both blood groups respectively (lanes 5 and 7).

**Figure 8 antibodies-13-00064-f008:**
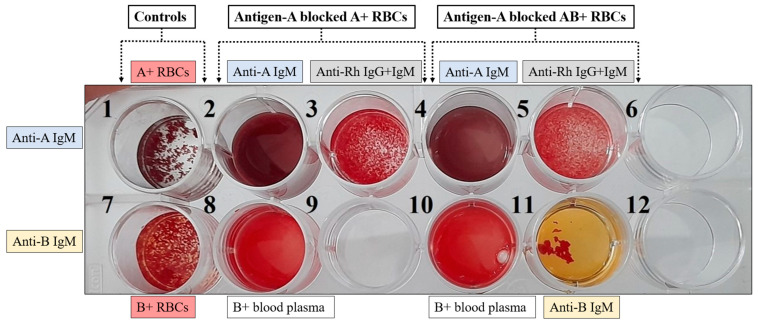
Blood incompatibility related hemagglutination reactions. Wells 1 and 7 show controls (A+ RBCs in anti-A IgM antibodies and B+ RBCs coated with anti-A scFv in anti-B IgM antibodies). Wells 2, 3 and 8 show antigen-A blocked A+ RBCs in anti-A IgM, anti-Rh (IgG and IgM), and B+ blood plasma containing anti-A antibodies respectively. Wells 4, 5, 10 and 11 show antigen-A blocked AB+ RBCs in anti-A IgM, anti-Rh (IgG and IgM), B+ blood plasma containing anti-A antibodies and anti-B IgM antibodies respectively. Wells 6, 9 and 12 are empty wells.

**Figure 9 antibodies-13-00064-f009:**
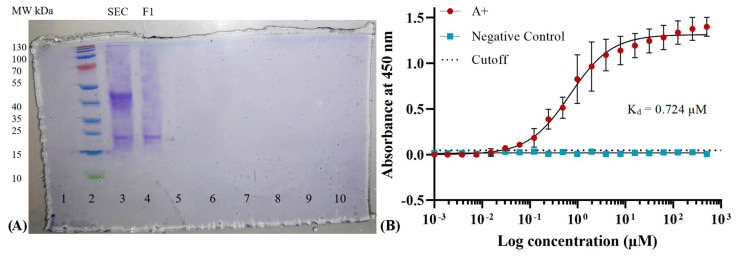
(**A**) Coomassie R-250 stained 15% SDS PAGE gel shows purification of functional anti-A scFvs attached on RBCs by Ni-NTA chromatography. Lane 3 shows SEC purified product and lane 4 (fraction F1) shows final eluted fraction from Ni-NTA chromatography. (**B**) Absorbance vs. Log concentration (μM) graph generated by GraphPad Prism shows dissociation constant K_D_ of monomeric functional anti-A scFv determined with a nonlinear regression model. Naked RBCs were used as negative control. Dilutions used ranged from 1 nM to 500 μM.

**Table 1 antibodies-13-00064-t001:** Molecular docking results of anti-A scFvs with antigen-A trisaccharide.

Single Chain Fragment Variable	Pose	Location	Interacting Saccharide	Binding Energies (kcal/mol)
Native Anti-A	1	Framework	Fuc, Gal, GalNAc	−6.0
	2	Framework	Fuc, GalNAc	−6.0
	3	Framework	Gal, GalNAc	−6.0
	4	CDR	Fuc, Gal, GalNAc	−5.7
	5	Framework	Fuc, Gal, GalNAc	−5.6
	6	CDR	Fuc, Gal, GalNAc	−5.6
	7	Framework	Fuc, Gal, GalNAc	−5.6
	8	Framework	Fuc, Gal	−5.6
	9	Framework	Fuc, GalNAc	−5.5
Mutant-1a Anti-A	1	CDR	Fuc, Gal, GalNAc	−7.4
	2	CDR	Fuc, Gal, GalNAc	−6.7
	3	CDR	Fuc, Gal	−6.7
	4	CDR	Fuc, Gal, GalNAc	−6.6
	5	CDR	Fuc, GalNAc	−6.3
	6	CDR	Gal, GalNAc	−6.2
	7	CDR	Fuc, Gal, GalNAc	−6.2
	8	Framework	Fuc, Gal, GalNAc	−5.9
	9	CDR	Gal	−5.8
Mutant-1b Anti-A	1	Framework	GalNAc	−6.2
	2	Framework	Fuc, Gal, GalNAc	−5.5
	3	CDR	Gal, GalNAc	−5.4
	4	Framework	Fuc, GalNAc	−5.4
	5	Framework	Fuc, Gal, GalNAc	−5.4
	6	Framework	Fuc, Gal	−5.4
	7	CDR	Fuc, Gal, GalNAc	−5.3
	8	Framework	Fuc, Gal	−5.2
	9	Framework	Fuc, GalNAc	−5.0

## Data Availability

Data will be made available on request.

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
