# Peer review of "Prevention of Blood Incompatibility Related Hemagglutination: Blocking of Antigen A on Red Blood Cells Using In Silico Designed Recombinant Anti-A scFv"

_2073-4468, 2024, doi:10.3390/antib13030064_

Round 1

Reviewer 1 Report (Previous Reviewer 1)

Comments and Suggestions for Authors

This article investigates the modification of ScFv targeting the surface antigen A on red blood cells (RBCs). The impact of introducing pairs of disulfide bonds at different positions on the stability of ScFv and antibody-antigen complexes was assessed using molecular dynamics simulations. The scarcity of universal blood type/O blood presents a significant medical challenge. Variations in carbohydrate modifications on RBC surfaces determine blood types, and recent advancements in enzyme-based modification to produce type O blood have shown promise. Blocking type A antigens, as discussed in this article, appears to be an effective approach towards this goal.

Based on the findings of this study, the following comments are noted:

1. The article provides limited quantitative data on the efficiency of the scFv in blocking antigen A and preventing hemagglutination. More detailed results would strengthen the conclusions.

2. Purification of ScFv monomers using centrifugal filters may not be effective.

3. In designing the expression plasmid for anti-A ScFv, the authors included a TEV protease cleavage site and subsequent His-tag at the C-terminal end. Was the His-tag removed from recombinant anti-A ScFv during subsequent experiments?

4. The experiment involving blocking of A+ RBCs used a 1:2 ratio of RBCs to ScFv solution. However, specific concentrations or qualities of ScFv used were not indicated, and the role of monomeric ScFv in subsequent results raises questions. Could further quantification of RBC numbers and monomeric ScFv ratio establish saturation levels?

5. ScFv molecules are inherently unstable. Despite the addition of inter-domain disulfide bonds, can they be stored at room temperature?

 Other minor corrections:

 "E.coli" should be italicized as "E. coli".

Comments on the Quality of English Language

No Comments

Author Response

Reviewer 2 Report (New Reviewer)

Comments and Suggestions for Authors

Saleha Hafeez and Najam Us Sahar Sadaf Zaidi's manuscript, titled "Prevention of blood incompatibility related hemagglutination: Blocking of antigen A on red blood cells using in silico designed recombinant Anti-A scFv," discusses how modified anti-A single chain fragment variable (scFv) produced and purified from E. coli can block the antigen-A on red blood cells. This approach addresses critical blood shortages by creating antigen-blocked red blood cells that remain functional and prevent clumping. This could expand the usable blood supply by enabling these modified cells across different blood groups during emergencies. However, some issues in the manuscript need to be rectified before accepting the paper. Some suggestions are provided below to improve the manuscript quality.

1. Please provide an explanation of Aliquots 1 to 3 in Figure 6 within the figure caption or methods section.

2. To ensure consistency in quality, please confirm if monomeric antibodies contain intramolecular scrambled disulfide bonds and address consistent positioning in subsequent products.

3. Did the authors observe monomer antibodies transforming into dimers or polymers during long-term storage? If so, how do the authors plan to prevent this from occurring?

4. Line 395: This study did not convert group A RBCs to other RBCs. It is recommended that the author modify this statement.

Author Response

Reviewer 3 Report (Previous Reviewer 3)

Comments and Suggestions for Authors

The authors have added experimental steps to further characterize the interesting antibody they use to block antigenic determinants on RBCs: first, they have added column filtration -based purification step, and second, they eluted the scFv bound to the RBCs and indeed, it appeared monomeric in SDS-PAGE analysis. They used this material to determine affinity of the scFv towards RBCs in ELISA. They have also included comments on the future prospects of the studied concept in the Discussion section, and improved Material and Method section with more details.

Figure 3D: Please check that the labels of the amino acids are correct. It appears that what is labelled Asp-50 looks like a Tyr, “Trp-33” is a small aliphatic amino acid, and so is “Tyr-182”, “Tyr-164” looks like a Trp residue.

Figure 9B: the authors show ELISA response of the monomeric scFv to the antigen. Please convert the lower axis into log units for better legibility of the results. What do error bars express (how many measurements were taken)? Please include negative control RBCs to assess background reactivity.

Line 178: “natural anti-A antibodies was incubated with antigen-blocked RBCs were incubated for 1 hour“ ; please correct

Author Response

This manuscript is a resubmission of an earlier submission. The following is a list of the peer review reports and author responses from that submission.

Round 1

Reviewer 1 Report

Comments and Suggestions for Authors

This article focused on the modification of ScFv targeting RBCs surface antigen A. The effect of introducing a pair of disulfide bonds at different positions on the stability of ScFv and antibody-antigen complexes was verified by combining them with molecular dynamics simulations. The shortage of universal blood type/O blood is indeed a difficult medical problem, different carbohydrate modifications on the surface of RBCs make differences in blood type, enzyme-based modification of red blood cells to produce type O blood has made many advances in recent years, and blocking of type A antigens as mentioned in this article seems to be an effective way to do this, so based on the content of this article, here are some of my comments:

1. In the experiment of blocking A+RBCs, a 1:2 ratio of RBCs to ScFv solution was used, but there is no indication of the specific concentration or quality of the ScFv used, and as far as the subsequent results are concerned it is the monomer ScFv that comes into play. So is it possible to quantify the number of RBCs and the ratio of the monomer ScFv further to determine the saturation ratio?

2. When designing the expression plasmid for the anti-A ScFv, the TEV protease cleavage site and subsequent His-tag were added at the C-terminal end, and did the authors remove the His-tag from the recombinant anti-AScFv during subsequent experiments?

3. Since size exclusion chromatography can not separate the monomers and dimers, how did the authors calculate the concentration of monomers purified after size exclusion chromatography?

4. In part 3.4 of the results, line 290, in terms of the results of figure 6, it should be Aliquot 2 displayed a minor decrease in ScFv concentration till 15 minutes for A+ RBCs and 10 minutes for AB+ RBCs. The authors have this part completely backward.

5. In part 3.5 of the results, the addition of anti-A antibodies did not cause the surface-bound ScFv of the A+ RBCs to fall off, indicating the high affinity of the single-chain antibodies. Then could the affinity of antigen A and native ScFv as well as mutant ScFv, anti-A antibody be measured to determine their affinity strength and to further confirm whether the introduction of disulfide bonds would have an effect on the affinity, and whether the effect of the introduction of disulfide bonds at different positions would make a difference?

6. ScFv are very unstable, can they be stored at room temperature even if inter-domain disulfide bonds are added?

Other minor errors:

1.      Page 2 line 75, “Escherichia. Coli” should be italic as “Escherichia. Coli”. The same error in other places, such as “E.coli”.

2.      Page 4 lines 151-152, the author used cellulose regenerated dialysis tubing (MWCO 8000-14000 kDa), I think the tubing should be 8000-14000 Da.

Comments on the Quality of English Language

Page 2 line 75, “Escherichia. Coli” should be italic as “Escherichia. Coli”. The same error in other places, such as “E.coli”.

Reviewer 2 Report

Comments and Suggestions for Authors

The authors produced anti-A ds-scFv in a bacterial system to block antigen-A of blood groups A and AB present in red blood cells to convert them to other groups and improve the blood demand in an emergency.

Anti-A scFv sequence was obtained from the Protein Data Bank and it was mutated to have an interdomain disulfide bond according to Zhao et al. (2010) [Reference 14]. Then bioinformatic analysis was performed to select a more stable mutant. The bacterial expression vector of anti-A scFv was acquired commercially and the antibody was produced and purified. After the affinity purification, 4 bands were observed and then size exclusion chromatography was performed without separation between dimers and monomers. As mutant scFv was selected in function of the structure stability, it was expected major production of monomers as shown by Zhao et al. (2010) and obtain monomers as the final product. Have the authors tried to improve the second purification? How would be the production of wild-type scFv?

The development of a standard assay would be important as proof of the concept and to reach the objective of the manuscript. For this goal, the development of a product with known concentration and high purity grade is recommended.

The manuscript's purpose is interesting. The authors should optimize the methodology to obtain a product with better quality by trying other experiment conditions. 

Reviewer 3 Report

Comments and Suggestions for Authors

In the present manuscript, the authors aimed at designing a stable antibody of scFv format to achieve specific blocking of blood group A-antigen, to render the RBCs of such type amenable for administration in transfusion to recipients with incompatible blood groups. They have designed a novel disulfide bond between the VH and VL of the scFv of interest in silico and performed extensive MD symulations, which show that the formation of the stabilizing motif is feasible, and docking experiments that show that the stabilized scFv indeed interacts with the desired trisaccharide antigen – in the designed stabilized mutant, the complexes with top negative binding energies indeed show interactions with the CDRs.

Further, they have expressed the scFv of interest in periplasm of E. coli and purified it via His-based affinity chromatography and gel filtration. They have then shown the depletion of this protein in blood plasma when incubated with A or AB-positive RBCs, and that blood samples treated with the scFv indeed ceased to agglutinate with anti-A IgM.

Regarding the translational importance of the study, which would mean extending availability of blood preparations for transfusion, the manuscript is a very important contribution towards improvement of human health. The article is well written and the reasoning behind the experiments clear. Nevertheless, the authors should explain certain points in experimentation:

-          There is no experimental evidence (except in silico) of actual stabilization of the scFv. Is there a possibility of these data be provided? There are also no data on experiments performed with the variant without the novel cysteine (either on expression, biophysical characterization or functionality)  – how can you argument that the mutation is really beneficial?

-          Could the dimerization of scFv that persists despite gel filtration point at reversible dimer formation? Regarding the fact that only the monomeric form is functional (i.e. depleted with the relevant RBCs), would an affinity chromatography step with the cognate antigen aid in purification?

-          The method of blocking the RBC antigen relies on very strong affinity of scFv – how can it be made sure that the antigen remains masked after the transfusion, and for how long can the masking persist?

Additionally, the order of Figures in appendix is not correct (please see the remarks below) and the bylines to the figures should be improved to be more informative.

Line 75: Escherichia. Coli: without full stop and the name of the species in italics (please correct throughout the text)

Line 115: Nco1 and Xho1: NcoI and XhoI

Line 133: “30 mins at 14,000 rpm” – for all centrifugation steps, please state the time, temperature and the speed should be in g units

Line 173, and throughout the text: sources of antibodies should be mentioned with RRIDs if available

Line 264: Figure A4 are i-tasser models

Line 193: Figure A5 is the photo of transformants on an agar plate. This Figure could also be omitted (does not add value).

Line 258: Figure A6 shows the absence of depletion of the scFv on B-positive blood;  reference to “refined views of MD simulations”, which is at present A7, is not mentioned in the text.

Line 282: Figure A7 is “refined MD simulations”; reference to graphs of purification outcomes (at present A8) is not mentioned in the text.

Line 296: “in supernatant” – bylines to Figures should be self-explanatory, please be more concise.

Line 375: Figure A8 shows purification outcomes.

Line 469: Figure A9: Possible structures of dimers made from monomers of anti-A scFvs – not mentioned in the text. The Figure byline should be more explanatory (where are these structures derived from and what does “CRD” in the Figure refer to?)

Round 2

Reviewer 2 Report

Comments and Suggestions for Authors

The authors answered partly the comments of the previous round.

They did not mention in the manuscript that other purification approaches were tried, and the results were not good. It is great to know that the group thought of other approaches to improve the purification step to obtain monomers. It is recommended to present these results in the present manuscript to publish a robust work that would be more interesting to the readers.

The strategy to select scFv with a stabilized structure was based on an in silico approach. Any assay, such as the binding assay to the target by ELISA, was not performed to confirm the results obtained by the in silico approach.

The “Recommendations for Authors” section received the same rate as the previous round because the authors did not mention if the second version of the manuscript presented improvements in the text and there are highlights without any explanation. That is one reason why the overall recommendation did not change.

The manuscript is not read to publication based on the comments above.

Reviewer 3 Report

Comments and Suggestions for Authors

In the revised version, the authors have corrected the order of the supplementary Figures, and improved the Figure titles. They also included responses to several questions in the “letter of response to the reviewer”. However, in my opinion these should be partly incorporated into the manuscript text, as they are important for the reader to judge on the potential translational value of the method. This in particular applies to the considerations on the required scFv antigen affinity. The authors have (in the response letter) extensively commented on possible outcomes of the alternative purification techniques including antigen-mediated purification: please indicate a potential protocol on how the experiments can be designed to obtain a monomeric form of the scFv, so that its basic biophysical characteristics could be determined in the first place (such as antigen affinity, thermal stability, and storage stability).

The last supplemental Figure should be designed in a way that the scheme and the molecular model match in colors.